# On the temporal spreading of the SARS-CoV-2

**Francesca Bertacchini[1]ᵒ, Eleonora Bilotta[2]ᵒ *, Pietro S. Pantano[2]ᵒ**

**1** Department of Mechanical, Energy and Management Engineering, University of Calabria, Rende, Cosenza, Italy, **2** Department of Physics, University of Calabria, Rende, Cosenza, Italy

ᵒ These authors contributed equally to this work.
* eleonora.bilotta@unical.it

**Data Availability Statement:** All relevant data are within the manuscript.

**Funding:** With regard to the ex 60% fund, indicated in the funding statement as "funds of the Physics Department of the University of Calabria", we

## Abstract

The behaviour of SARS-CoV-2 virus is certainly one of the most challenging in contemporary world. Although the mathematical modelling of the virus has made relevant contributions, the unpredictable behaviour of the virus is still not fully understood. To identify some aspects of the virus elusive behaviour, we focused on the temporal characteristics of its course. We have analysed the latency trends the virus has realized worldwide, the outbreak of the hot spots, and the decreasing trends of the pandemic. We found that the spatio-temporal pandemic dynamics shows a complex behaviour. As with physical systems, these changes in the pandemic's course, which we have called *transitional stages of contagion*, highlight shared characteristics in many countries. The main results of this work is that the pandemic progression rhythms have been clearly identified for each country, providing the processes and the stages at which the virus develops, thus giving important information on the activation of containment and control measures.

## 1. Introduction

Since the first confirmed case in China, the novel Coronavirus disease 2019 (COVID-19) [1] caused by SARS-CoV-2 virus, has raised great concern around the world. Whilst a lot of attention has been given to understand the virus effects on people, due to the huge burden of human mortality and morbidity caused by it, less attention has been paid to explaining the spatio-temporal variability of the virus behaviour worldwide. Furthermore, as it is already known, hot spots all over the world have arisen in different time, giving the idea that there is a sort of evolutionary strategy of this novel RNA based Corona Virus [2, 3]. Some authors have already investigated the infectious disease ecology, from the first mathematical modelling of infectious diseases of Bernoulli [4], to the Nobel Laureate Ross [5], to Kermack and McKendrick [6, 7], just to cite a few examples. Some of them finding interesting correlations with power law expression [8, 9], also for the Covid-19 disease [10, 11]. Although we have models of the major currently known infectious diseases [12], such as HIV [13], malaria [14], SARS-coronavirus [15], rabies [16], and influenza [17], the unpredictable behaviour of SARS-CoV-2 has been unexpected. Holmes [18] reports that RNA viruses evolution are due to their intrinsically high rates of mutation, on the contrary of most DNA-based organisms that produce higher replication fidelities copies. Additionally, the extreme mortality and morbidity caused by the

indicate that the funds are: For BE: 2016.RIC.BILO.
ATE001 FRS, UNIVERSITY OF CALABRIA
Modelling and Applications, University Projects.
For PSP: 2016.RIC.PANT.ATE001 FRS
UNIVERSITY OF CALABRIA, Modelling, University
Projects. There was no additional external funding
received for this study.

**Competing interests:** The authors have declared
that no competing interests exist.

SARS-CoV-2 virus is due to its relatively controlled genome sizes, that, in order to survive must rely on a combination of proliferating mutations and huge population size to create the genetic diversity it needs to adapt to new environments in different times [19, 20]. For this reason, documenting the mechanism of evolutionary change may be critical to the design of future intervention strategies. To account for these dynamics, we hypothesized that, given the magnitude of the phenomenon and the genetic drift of the virus, more sophisticated models than traditional ones intercept the virus self-replication process, and therefore the evolutionary strategies to spread in all the environments of the earth.

Let us assume that the self-replicating behaviour of the virus needs space in order to achieve the pandemic dynamic that has now claimed millions of infected and hundreds of thousands of deaths worldwide. In manifesting its infectious power, giving rise to several outbreaks in the world, the virus used different times. That is to say, that, depending on the countries in which the virus has travelled, different times of propagation have occurred. In this article, we show that the virus behaviour has scale invariance, considering the number of infected for each country and for all countries in the world. We think that propagation in space and time of the SARS-CoV-2 follows a power law and that the phenomenon has a fractal nature [21, 22]. Moreover, we show that the pandemic phenomenon can be compared to typical behaviour of phase transitions that captures chaotic dynamics [23, 24].

The article is organised as follows. After this introduction, Sections 2 and 3 deal respectively with the spatial and temporal spread of the virus. Section 4 illustrates the transition phases of the virus evolution. The used methods are given in Section 5. Conclusions close the work.

## 2. Geo-spatial diffusion of epidemic

As our main interest in this paper is to put in evidence the temporal aspects of the Covid-19 evolution, we synthetize the spatial analysis in Fig 1. The main routes of contagion seem to spread from China to Continental Europe, and then to USA, UK and Russia, even though, other routes are going to be realized especially in the Southern Hemisphere. We plot the total number of confirmed cases per day, beginning with January 22, 2020, according to the WHO reports on this pandemic [25]. By ordering the countries according to the number of infected and numbering them progressively from 1 to 217, we get the infected country rank order.

As it is possible to see, Fig 1(a) shows the countries that have not been infected, listing the names of these countries. For the most part, they are isolated small islands or countries from which no official data comes from. Fig 1(b) and 1(c) show countries with more than 10,000 and 100,000 of infected respectively. The spatial distribution of the infected shows the extreme heterogeneity of the population both from a genetic and behavioural point of view, which could also correspond to the genetic differentiation of the virus (Forster, 2020). Furthermore, it seems that in the first phase of its evolution, the virus privileges the 32nd and 52nd parallel. This is the latitude of countries with more than 10,000 infected. The 50 most infected countries (with more than 100,000 infected) are mainly around the 50th and the 20th parallels. Fig 1(d) shows the direction of the virus, ideally represented by arrows, while spreading causing hot spots in different countries. Fig 1(e) shows the population number versus the number of infected on a logarithmic scale. In the red circle, some of the countries that have been most infected. As several authors have already discussed [10, 11, 26], we think that the spatial diffusion of Covid-19 follows a power law. In fact, the plots in Fig 1(f) is calculated on a logarithmic scale, representing the number of infected people for the 217 countries and a ship, while the red curve of Fig 1(g) shows how a straight line intercepts this logarithmic distribution. Fig 1(h) plots how the previously obtained exponential function fits real data. We have connected in a graph the countries in order to obtain a visual representation of the connected countries by

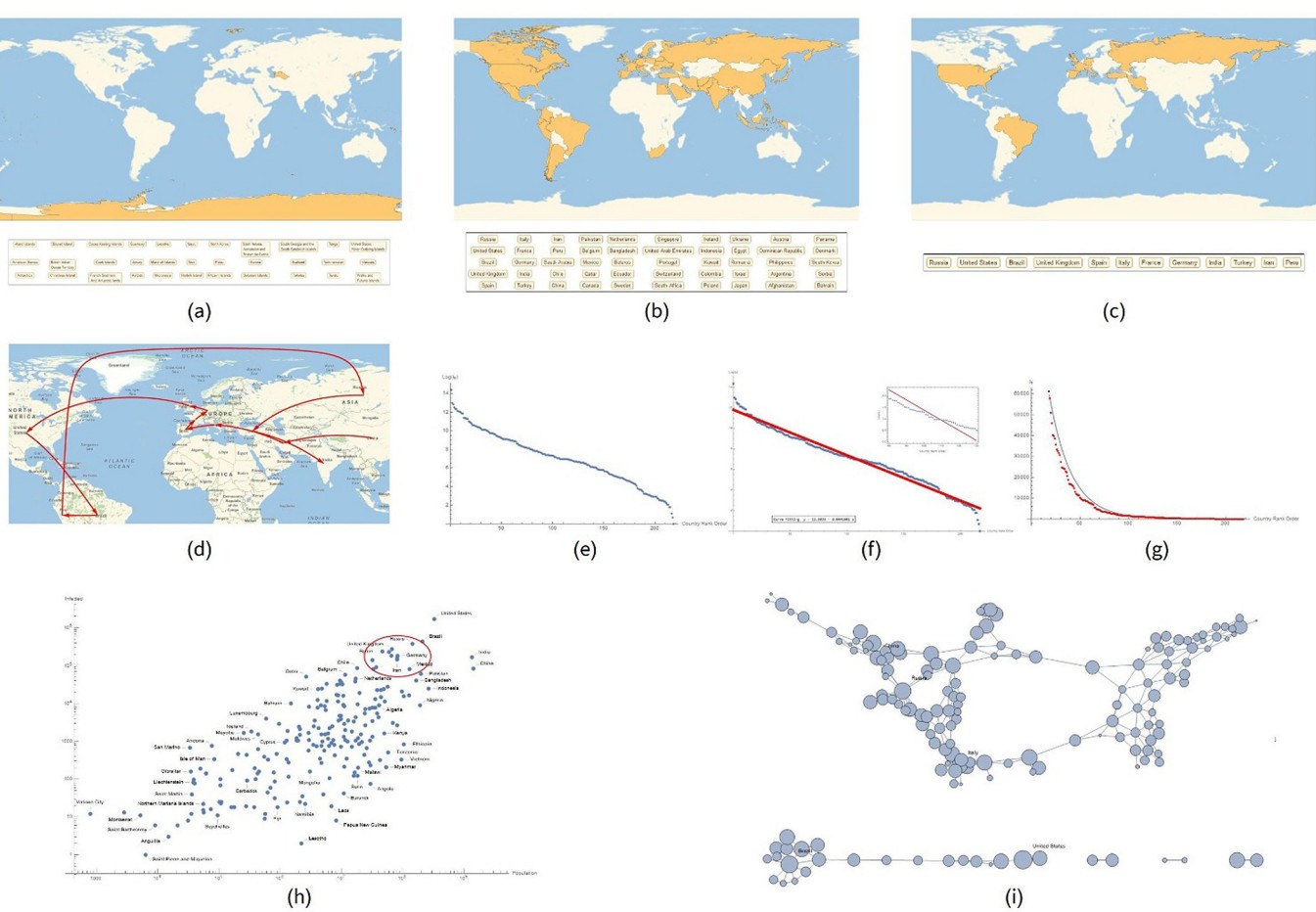

**Fig 1. Geo-spatial transmission of the SARS-CoV-2.** In this figure, (a), (b) and (c) represent the Countries of the world with almost no infected, 10,000 and 100,000 number of infected respectively. (d) Presents the spatial ignition of each country, represented as red arrows. (e) Shows the exponential function fitted with real data. Some of the most infected countries are in the red circle. (f), (g) and (h) represent the total number of infected people per country, the Log plot on a logarithmic scale, and the population number for each country and the number of infected respectively. Finally, (i) shows a graph of the 217 countries with some of the most infected ones. In the graph, the nodes represent countries while the links connect geographically neighbouring countries. The data collection began on January 22, 2020 and ended on May 28, 2020.

the virus infection. In the graph of the 217 infected nations, the edges of the network connect geographically neighbouring countries. To these edges, other links between China and the countries that were infected in the first 15 days have been added (Fig 1(i)). In red are highlighted China, Russia, Brazil, Italy, and the US.

## 3. Covid-19 temporal evolution follows an exponential law

To consider the temporal dimension of the virus behaviour, we have collected all starting times for the SARS-CoV-2 for all the countries. In order to account for these temporal dynamics, we have therefore tried to extract from the complexity of the pandemic phenomena some temporal variables, which can be helpful for a deep understanding of the space-time behaviour of the virus. Starting from the conventional date of the beginning of the pandemic in China, (January 22, 2020), we have tracked the time of virus ignition in every country in the world. This allowed us to have a space-time map of the virus and to notice how each country had different starting times of the pandemic. Then we analysed what we called the phase periods of

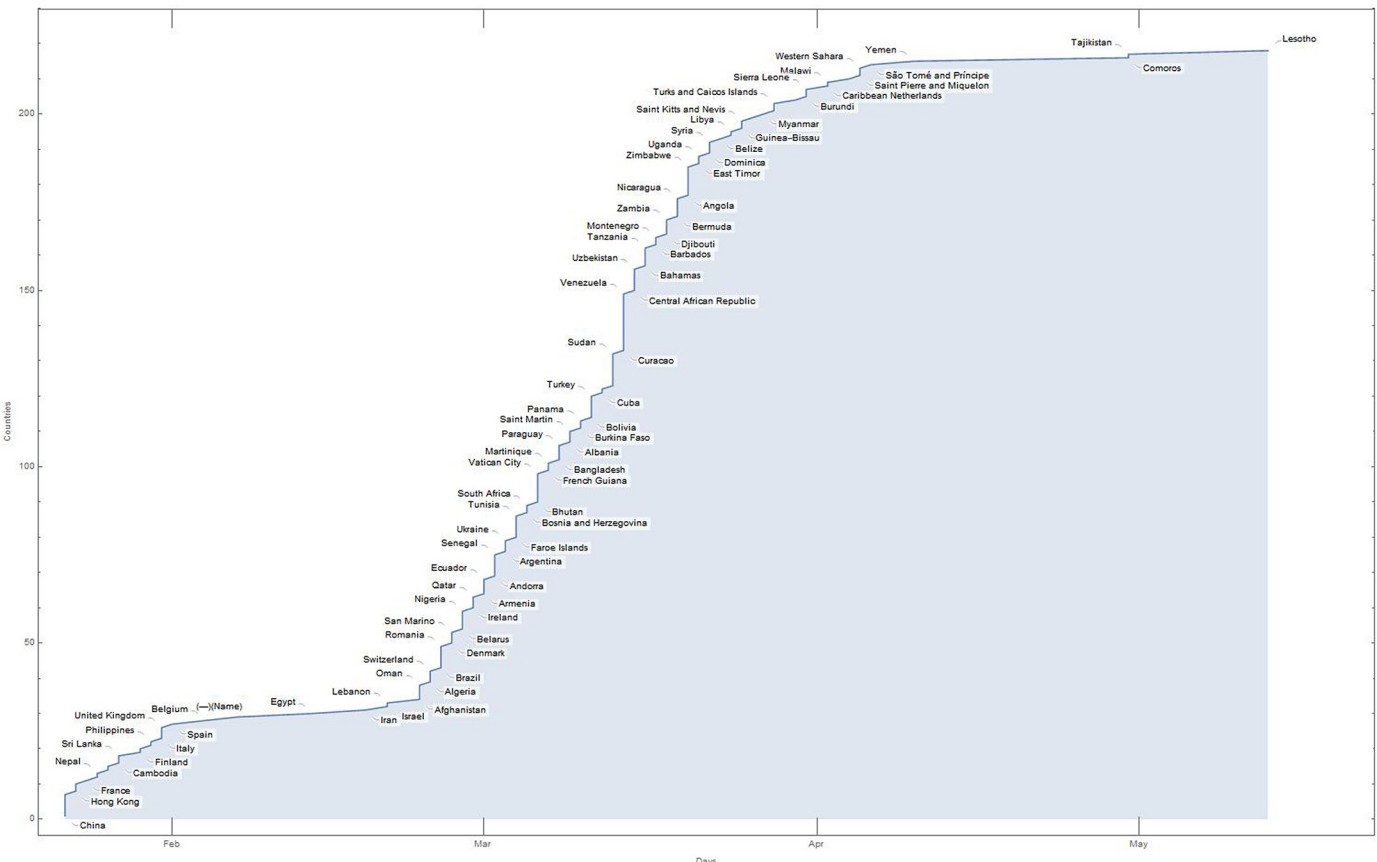

**Fig 2. Starting times for each of the 217 countries of the world.** The days of the beginning of the pandemic, from January 22, 2020 to May 28, 2020 with respect to countries.

the virus evolution, identifying a conventional metric based on the pandemic growth from 1 to 10 to 100 to 1000, to 10000 to 100000 number of infected people. This metric (and the choice to calculate only the number of infected) is purely conventional. We could have used any variable of the classical compartmentalised models. What interests us is whether there are general patterns of virus behaviour, through the analysis of the effects that the virus produces when it completes its viral expansion. In this sense, the data are simply indirect representations of the virus behaviour.

A display of the starting times for each of the 217 countries of the world is shown in Fig 2.

On the first day of the pandemics (conventionally set on the 22 January 2020 when China declares the existence of an unknown virus that causes several adverse effects in humans and even death), official cases were reported in seven countries. The rapid spread of the virus and its diffusion through different human travelling systems [27–29], amplified the phenomenon of contagion. After 10 days, the number of countries, with at least one case of Covid-19, had become 26, showing the strong ability of the virus to penetrate the different nations of the world. This is quite evident from Fig 2, with a rapid growth in the number of involved countries, as time progresses. On the tenth to the twentieth day, there is a slowdown in the penetration of the pandemic, with only five new countries included in the pandemic list. This causes a flattening of the curve. From the 30th day, the penetration of the pandemic resumes with a very rapid growth that increases until the 65th day. At this time, the plague had reached 195

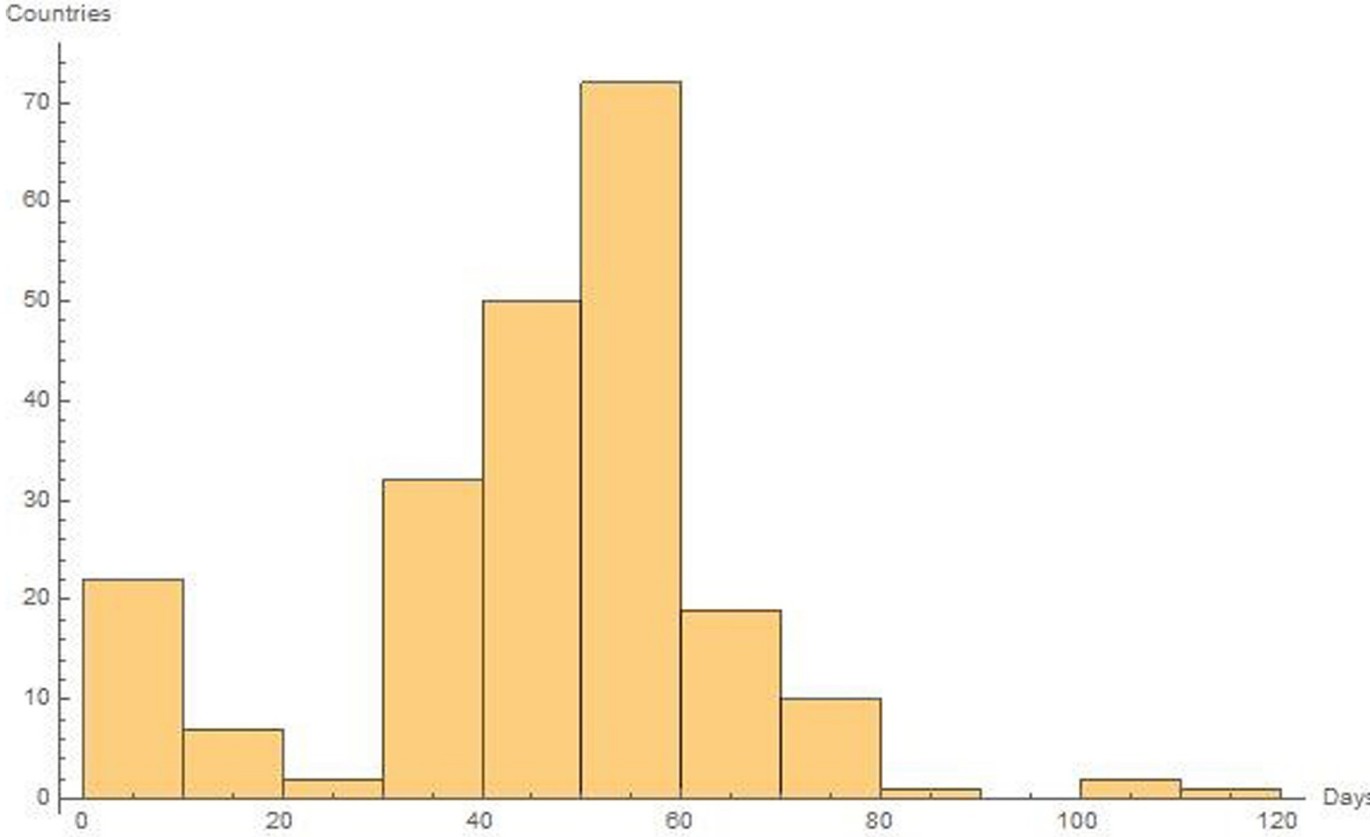

**Fig 3. Twofold trends in the development of the pandemic all over the world.** As it is possible to see, after an initial stasis of the contagion, as the days progress, the pandemic spread significantly continues and then decays.

countries and then, luckily, it begins to decline. (S1 File). A picture of these twofold trends in the development of the pandemic is evident in Fig 3.

If we analyse the distribution of the number of infected people for each country over time, we observe that pandemic evolution follows an exponential law. Starting from the day of the beginning of the pandemic for each country, we have placed all the countries on the x-axis. Then, we assigned a number to each country and rearranged the Country Rank Order according to time. Fig 4 shows this trend on a logarithmic scale.

Therefore, the distribution of the contagion in the world is not at random, but it follows a precise configuration.

## 4. The transition phases of the virus evolution

In order to analyse the most significant elements of the SARS-CoV-2 dynamics, we have considered the following aspects:

1. The official confirmation of almost one infected individual in each country, for all countries;

2. The hot spots ignition all over the world;

3. The slowdown of the pandemic curve.

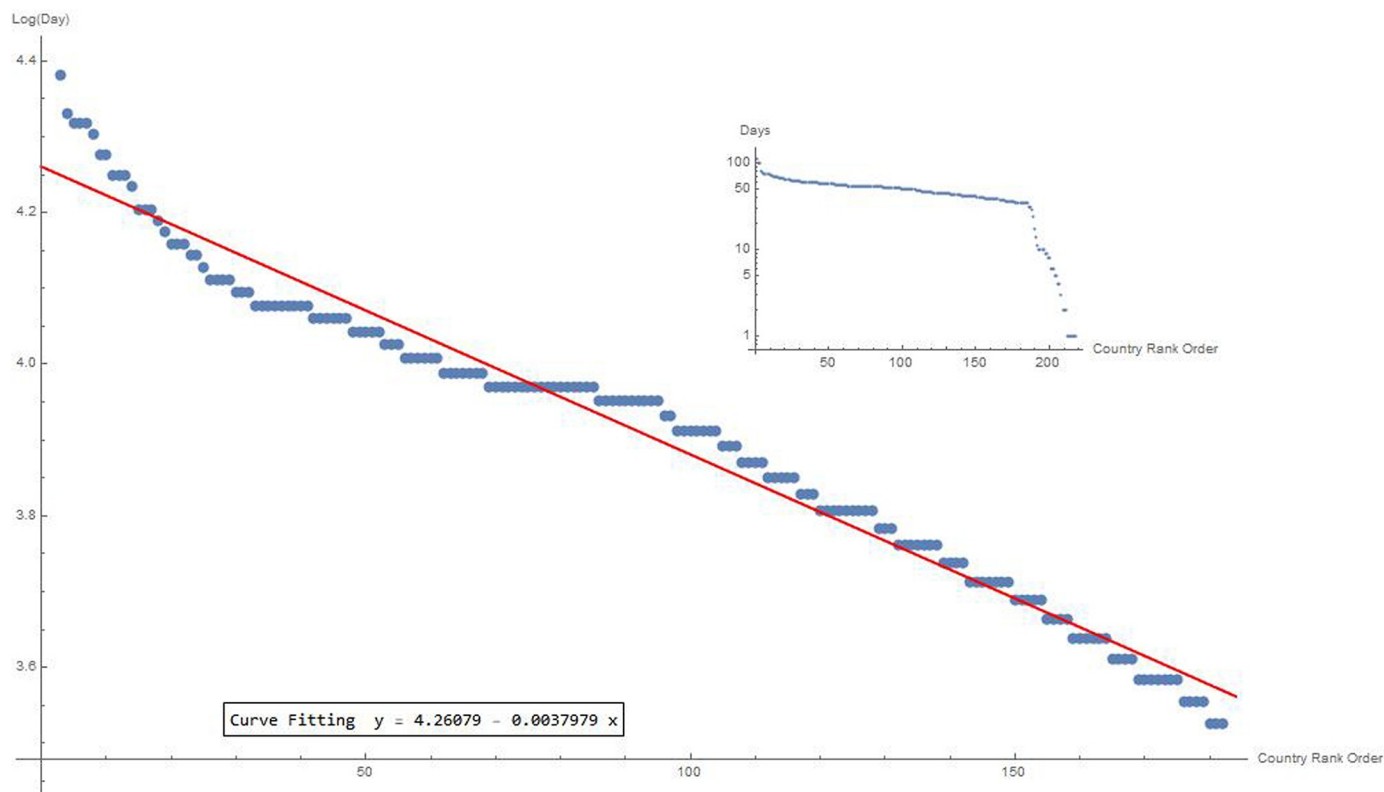

**Fig 4. Starting day distribution for each country over time.** Realized on a logarithmic scale, this distribution intercepts an exponential law dynamics. The regression curve value is y = 4.26–0.0038x.

What we have found is that not only the pandemic ignition times structure themselves in an articulated way over time, but correspondingly, the growth of the contagion increases over time with peculiar characteristics for each nation (Fig 5).

In the plot, a point on the x-axis represents each country with at least 10 cases on the day of the first case and on the y-axis on the day of the tenth confirmed case. The red line represents the nations having both the first and the tenth case on the same day. The green line defines the countries that took a week to reach the tenth case. Nations progressing from 1 case to 10 cases are placed between the two lines. To make this trend more explicit, let us consider the day on which each single country changes the scale from 1 to 10 to 100 to 1,000 to 10,000 to 100,000 to 1,000,000. Table 1 reports these values for 44 countries, with at least 10,000 cases recorded on May 28, 2020. (S2 File). If we sort the time differences between the first and the tenth case, we observe that the curve seems to have an exponential trend that therefore can be captured by a straight line on a logarithmic scale. Unlike the country rank order concept adopted to illustrate the spatial distribution of the infected in the world, in this case we made an interval rank order on the day the pandemic started for each country with almost 10 infected individuals. Starting from the day of the beginning of the pandemic for each country, we have placed all the countries on the time axis. Then we have assigned a number to each country and rearranged the countries according to the time parameter. This trend can also be intercepted by a Zipf distribution (Fig 6), usually given in a logarithmic scale. In particular, Zipf argued that some events are distributed linearly, because they derive from a power law. If this happens, then we have a potential fractal structure [2–22]. Other more sophisticated analyses are required to ascertain the fractal nature of the phenomena under observation.

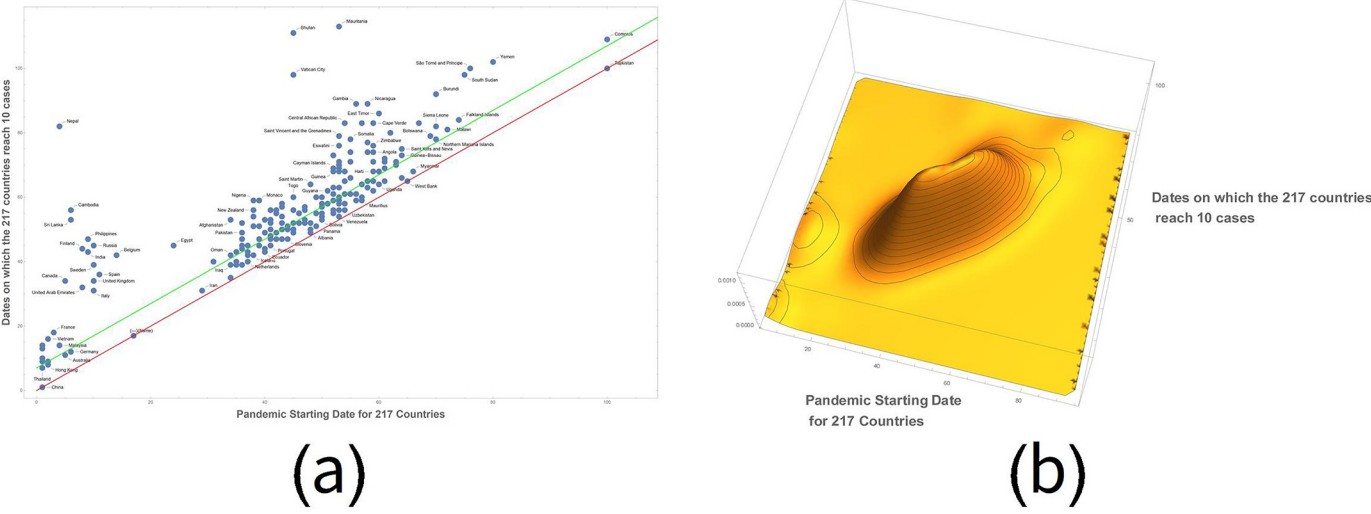

**Fig 5. Plot of the countries that reached the tenth case, from the official onset.** In (a), 67 nations got this time and are therefore included in the space between the red line and the green line. In the plot, on the x-axis, the day on which the first case is recorded and on the y-axis the day on which the first 10 cases occurred. In (b), a 3D plot of the process. It is possible to note the Chinese evolution of the pandemic (left), the subsequent European phase with the pandemic explosion in Italy and Spain, later followed by the contagion of the US countries.

Pandemic progression of the 44 countries passing from 1, 10, 100, 1000, 10000, 100000 infected individuals are listed in Table 1.

By analysing these times further, we realize that even in this case, the distribution follows trends, which are also repeated in the time intervals from one scale to another (Table 2).

## 4.1 Switch-on time of hot spots around the world

In many countries of the world, the day on which the first infected person is first recorded is not automatically the day on which the infection begins to grow. For example, in Italy, the first infected people are registered on the 1st February 2020, the 10th day since the official start of the pandemic in China. The couple of Chinese tourist coming from Wuhan do not seem to have infected anyone. Therefore, no hot spot is due to these cases. The starting date of the hot spot in Italy can be considered the 30th day from the beginning of the pandemic in China (22 February 2020), when the number of registered cases is attested to 10. From this date, the number of infected increases from 10 to 100 cases in two consecutive days (first very fast change of scale) and from 100 to 1000 in the following 6 days (second change of scale). If we consider the time intervals to go from one order of magnitude to another for each country, we can define the starting time of the hot spot for a country as the day to which corresponds the smallest interval of the pandemic growth. As with physical systems, these are significant changes to the pandemic's course, which we have called transitional stages of contagion. These changes highlight an increase in the threat of contagion and viral transmission. See data on the hot spot explosion times (S3 File) and the smallest intervals from the starting time of the hot spot for each country and the hot spot explosion (S4 File).

## 4.2. The latency period

From the analysis of the virus evolution times in each country, we have noticed that before the actual contagion begins and follows its temporal development, the virus has latency times, different for each part of the world. We define the latency time as the difference in time between the first case recorded in a country and the hot spot outbreak. This makes us think that the

**Table 1. Pandemic progression.**

| Country | Inf. = 1 | Inf. = 10 | Inf. = 100 | Inf. = 1,000 | Inf. = 10,000 | Inf. = 100,000 | Inf. = 1,000,000 |
|---|---|---|---|---|---|---|---|
| Russia | 10 | 45 | 56 | 66 | 79 | 100 | x |
| United States | 1 | 13 | 42 | 50 | 58 | 66 | 98 |
| Brazil | 36 | 45 | 52 | 60 | 74 | 103 | x |
| United Kingdom | 10 | 34 | 44 | 53 | 65 | 86 | x |
| Spain | 11 | 36 | 41 | 48 | 56 | 71 | x |
| Italy | 10 | 31 | 33 | 39 | 49 | 69 | x |
| France | 3 | 18 | 39 | 47 | 58 | 77 | x |
| Germany | 6 | 12 | 40 | 47 | 57 | 75 | x |
| India | 9 | 43 | 53 | 68 | 83 | 118 | x |
| Turkey | 50 | 55 | 58 | 61 | 69 | 93 | x |
| Iran | 29 | 31 | 36 | 41 | 51 | 106 | x |
| Peru | 45 | 49 | 56 | 70 | 84 | 120 | x |
| Saudi Arabia | 41 | 47 | 53 | 65 | 90 | x | x |
| Chile | 42 | 49 | 55 | 64 | 89 | x | x |
| China | 1 | 1 | 1 | 4 | 11 | x | x |
| Pakistan | 36 | 49 | 55 | 64 | 92 | x | x |
| Belgium | 14 | 42 | 45 | 55 | 68 | x | x |
| Mexico | 38 | 51 | 58 | 70 | 93 | x | x |
| Qatar | 39 | 47 | 50 | 73 | 96 | x | x |
| Canada | 5 | 34 | 50 | 60 | 72 | x | x |
| Netherlands | 37 | 40 | 45 | 54 | 68 | x | x |
| Bangladesh | 47 | 56 | 76 | 84 | 104 | x | x |
| Belarus | 38 | 51 | 69 | 78 | 96 | x | x |
| Ecuador | 40 | 43 | 57 | 63 | 90 | x | x |
| Sweden | 10 | 39 | 45 | 54 | 81 | x | x |
| Singapore | 2 | 9 | 39 | 71 | 92 | x | x |
| United Arab Emirates | 8 | 32 | 57 | 72 | 96 | x | x |
| Portugal | 41 | 45 | 52 | 59 | 74 | x | x |
| Switzerland | 35 | 39 | 44 | 52 | 64 | x | x |
| South Africa | 44 | 50 | 57 | 66 | 110 | x | x |
| Ireland | 39 | 45 | 53 | 62 | 83 | x | x |
| Indonesia | 41 | 48 | 54 | 66 | 100 | x | x |
| Kuwait | 34 | 35 | 53 | 81 | 112 | x | x |
| Colombia | 45 | 52 | 58 | 71 | 108 | x | x |
| Poland | 43 | 47 | 53 | 64 | 92 | x | x |
| Ukraine | 42 | 56 | 64 | 73 | 100 | x | x |
| Egypt | 24 | 45 | 53 | 74 | 112 | x | x |
| Romania | 36 | 47 | 53 | 65 | 93 | x | x |
| Israel | 31 | 40 | 50 | 61 | 80 | x | x |
| Japan | 1 | 9 | 31 | 60 | 88 | x | x |
| Austria | 35 | 40 | 47 | 55 | 70 | x | x |
| Dominican Republic | 40 | 53 | 60 | 70 | 110 | x | x |
| Philippines | 9 | 47 | 53 | 67 | 106 | x | x |
| Argentina | 42 | 47 | 59 | 70 | 122 | x | x |
| Afghanistan | 34 | 53 | 66 | 90 | 124 | x | x |
| Panama | 49 | 51 | 58 | 70 | 121 | x | x |
| Denmark | 37 | 43 | 49 | 57 | 107 | x | x |

(*Continued*)

**Table 1.** (Continued)

| Country | Inf. = 1 | Inf. = 10 | Inf. = 100 | Inf. = 1,000 | Inf. = 10,000 | Inf. = 100,000 | Inf. = 1,000,000 |
|---|---|---|---|---|---|---|---|
| South Korea | 1 | 10 | 30 | 36 | 73 | x | x |
| Serbia | 45 | 50 | 58 | 71 | 109 | x | x |
| Bahrain | 34 | 35 | 49 | 81 | 128 | x | x |

Table 1 shows the days when countries officially have 1,10,100,1000, 10000, 100000 infected on the rows. We indicated with 1 the starting day that corresponds to January 22, 2020 and with the following numbers to indicate the days progression. 50 countries passed from 1 to 100,000 infected. Only one country has increased to 1 million of infected.

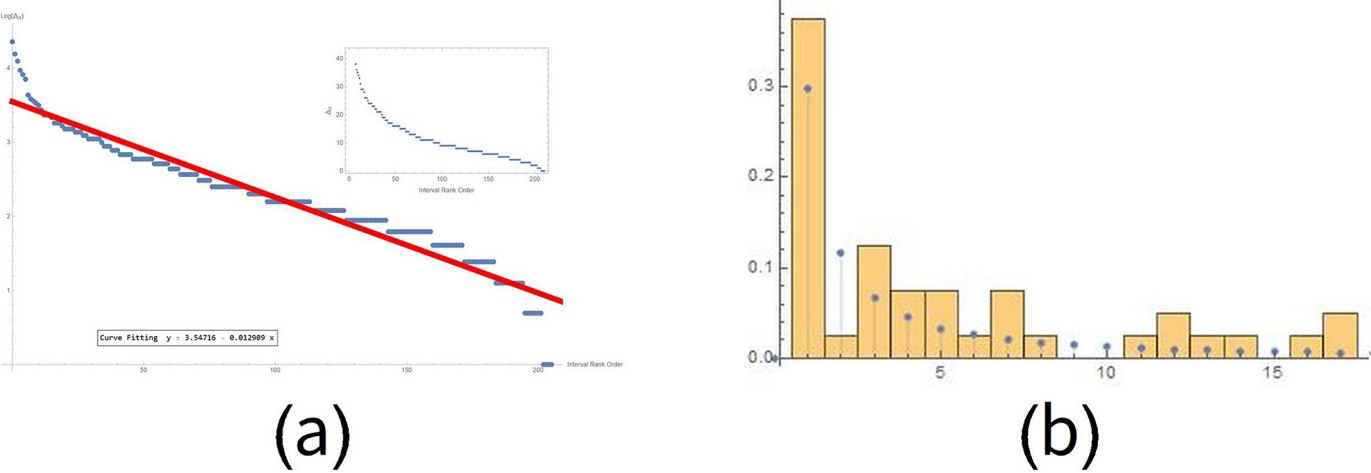

**Fig 6. Time differences between the first and the tenth case for the countries.** If we order all the intervals $i$ according to $\Delta_i$ values and number them progressively from 1 to 217, we get the interval rank order, curve fitting $3.548 - 0.013x$ (a). Furthermore, the figure compares the distribution histograms considering the real data (with full rectangles) and those obtained from a Zipf distribution (with points) (b).

virus began to circulate in a nation long before the hot spot broke out, but its adaptation processes took time for the pandemic to start manifesting itself in a disruptive manner. For each country, there are divergent studies in relation to the precise time at which the virus began to circulate. Therefore, it seems reasonable to start measuring this starting time unambiguously from the time when the first case is recorded. Fig 7 shows this latency times for the 44 countries with more than 50 recorded cases.

**Table 2. Interpolation lines.**

| Time intervals in changing scale | Interpolating lines |
|---|---|
| 1->10 | $3.548 - 0.013x$ |
| 10->100 | $3.743 - 0.0143x$ |
| 100->1000 | $3.907 - 0.0194x$ |
| 1000->10000 | $3.970 - 0.0380x$ |

On the left column, time intervals in the number of infected individual. On the right column, the fitting curve values for countries passing through the progression that goes from 1 to 10, to 100, to 1000, to 10000, to 100000 and so on. The data fit is performed on a scale (x, Log(y)). Interestingly, the slope of the interpolating curve increases from one interval to another.

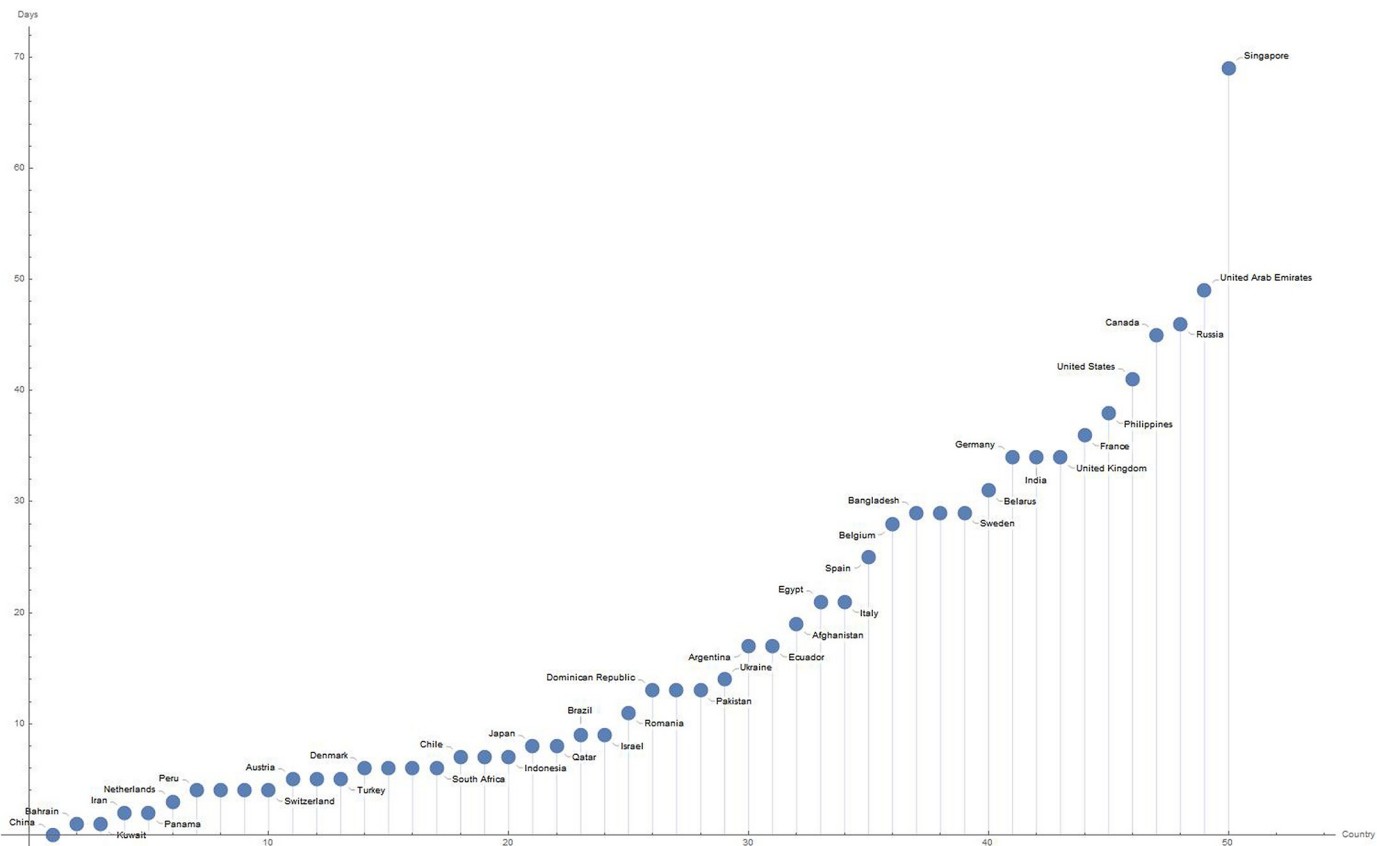

**Fig 7. Latency times for 50 countries with more than 10000 recorded case.** The plot displays the countries on the x-axis and the days of latency on the y-axis, considering as starting day, January 22, 2020.

The latency time explains why in countries like France and Germany, although the virus has been detected before than in Italy, it took longer for the hot spot to follow than in Italy. By analysing the pandemic trend, we observe an epicentre in the first phase in China and then in Italy/Spain and then in the USA, etc. Actually, the latency time provides us with an important information to understand the virus behaviour that can be further explored. In Germany, France, United States etc. the hot spots occur not when the 10th case is registered but when the 100th case is registered, while in Singapore the hot spot starts from the 1,000th recorded case.

## 4.3 Turning points allows us to define how effective the reaction of each country has been useful

It is well known that logistic maps can be useful to interpret analytically the course of a pandemic. While these models can present many errors in forecasting the phase of the epidemic trend, especially in the early stages when the number of data is insufficient, they become excellent representations of the trend of the phenomenon. To thicken the data, we have used an adaptation of the logistic function (https://community.wolfram.com/groups/-/m/t/1899911):

$$d_n(t) = \frac{A}{1 + e^{\frac{-t + t0}{T}}}$$

where $d_n(t)$ represents the distribution of the infected referred to the nation $n$ at time $t$ and $A$, $t0$ and $T$ are three constants determined by the fitting between the function and the real data.

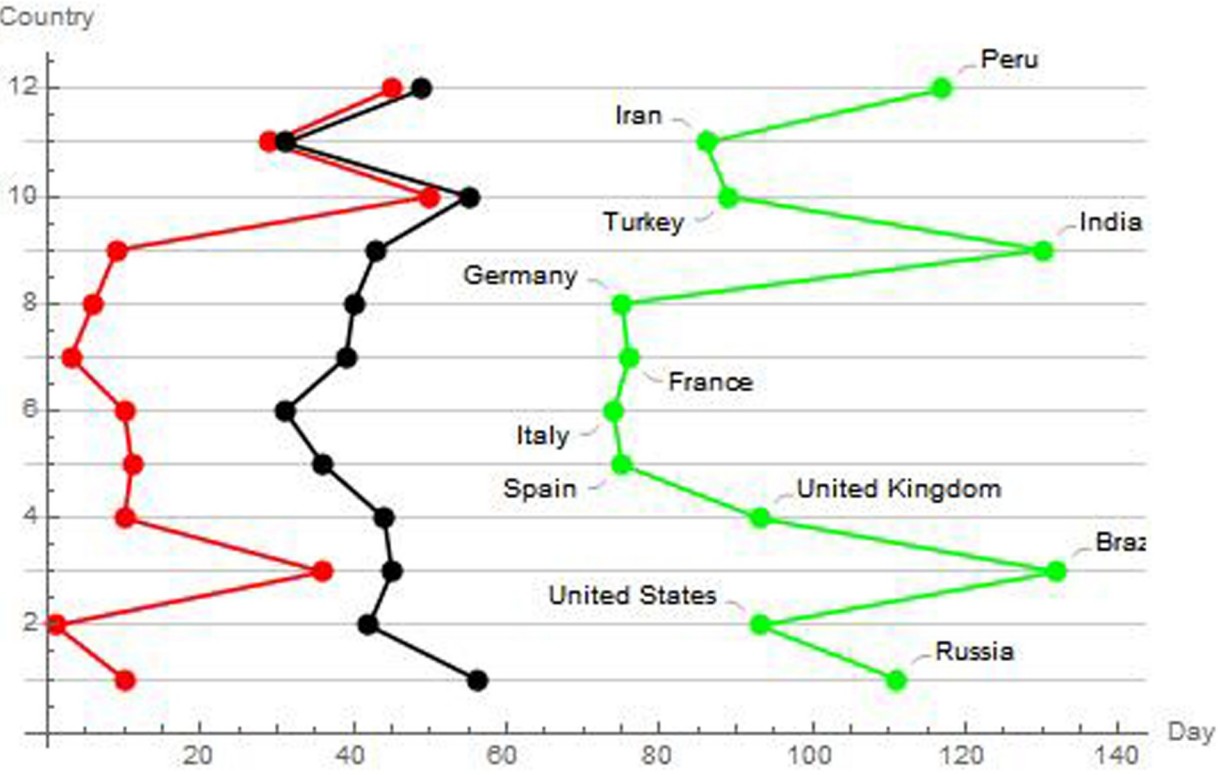

**Fig 8. Trends of the rhythms of evolution for the starting dates (red), hot spot expansion dates (black) and turning points of the pandemic dates (green) for the 12 most affected countries.** The image has been obtained by considering the intervals among these observation points, and by sketching on a plane the differences, obtaining the temporal intervals of the pandemic course. In analogy with physical systems, we have called these intervals as phase changes in the pandemic trend.

$t0$ represents exactly the turning point, that is when virulence changes its nature by decreasing the daily number of infected. The results data are reported in the attached excel file (S5 File).

It may be interesting to observe the temporal relationship among the pandemic beginning, the hot spots ignition and the turning points starting points. Subtly, this connexion gives us a measure of the effectiveness of the containment measures. By considering the intervals among these observation points, we sketched on a plane the differences, obtaining the temporal intervals of the pandemic course. In analogy with physical systems, we have called these intervals as *phase changes* in the pandemic trend. Consequently, it is quite clear from the Fig 9, relating to the 12 most affected countries, such as the United States, Brazil and Russia have not taken effective action. A finer analysis also highlights the small differences between France and Germany, where a better intervention than the latter reduced their overall rates of infected, compared to the former. It is also clear the slight delay in the effectiveness of Italy's containment measures, compared to these other countries (Fig 8).

## 4.4 Extracting rhythms from the temporal spreading of the contagion

The question now is whether the dynamics of the virus follow predefined patterns. As we have seen previously, it can be interesting to investigate the days between changes in scale. This data, already used previously, is reported in the attached excel file (S4 File). In the following Fig we report these changes for the 12 most infected countries, on an absolute time scale

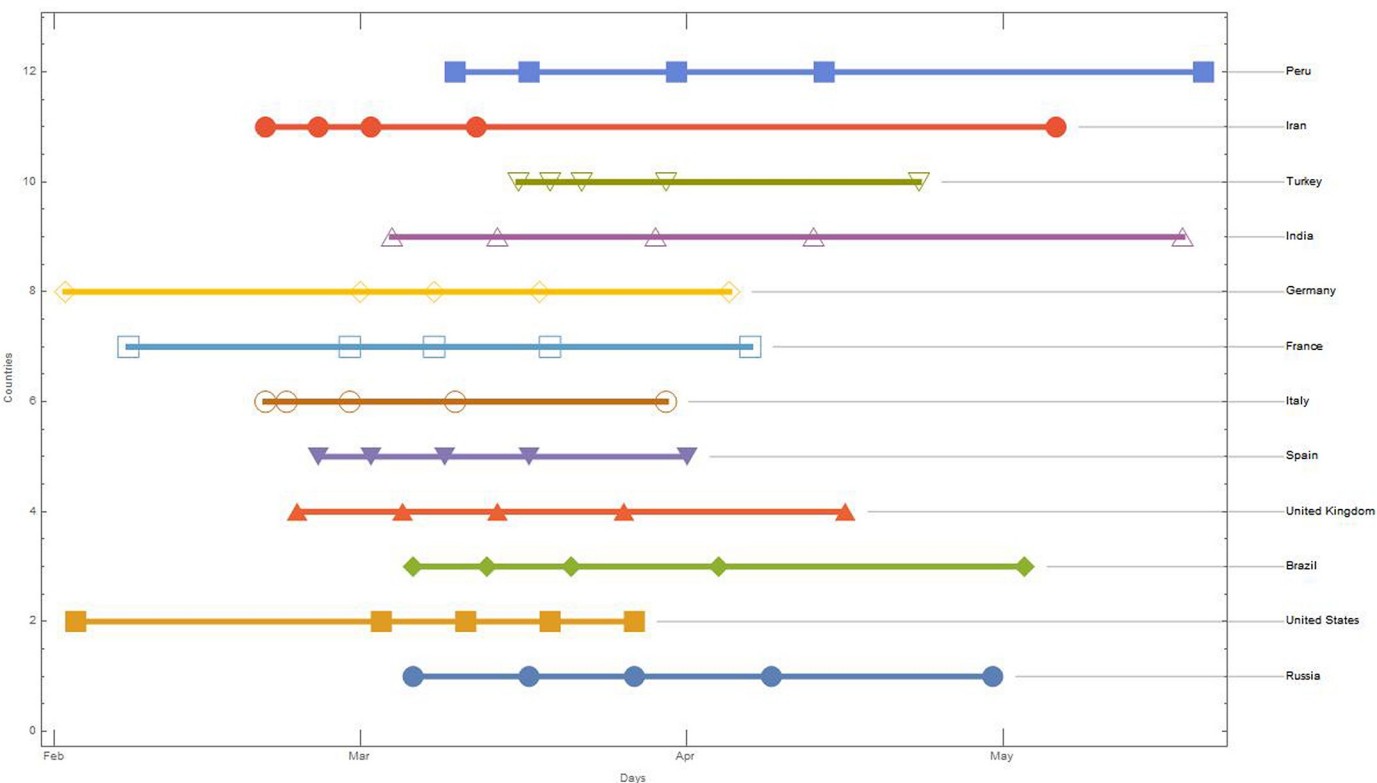

**Fig 9. Scale change days for countries with more than 100,000 infected people.** The shapes on the line represents these temporal changes for each country.

starting from day 1 (22 January 2020). In Fig 9, we report the scale change days for countries with more than 100,000 infected people.

As we have already said, we can identify a day when the hot spot will start, considering the previous period as the time of incubation/adaptation of the virus. In this phase, we are interested in the growth rate of the virus, starting from the hot spot and the subsequent behaviour after the lockdown. This then leads us to select only the scale change intervals from the hot spot. We have calculated the steps between the points of interest that we have considered essential to analyse the temporal trend of the virus in each country (beginning, hot spot ignition, turning point of the pandemic trend) by counting the differences between the days in the temporal succession that identify these evolutionary points. For the 12 countries with more than 100,000 infected, the result is shown in Table 3.

For some countries like USA, Spain, Brazil, Italy, Turkey, India, Peru and Iran, we can identify four intervals while for the others we will have only three intervals. If we instead analyse the 50 countries with more than 10,000 infected, the number of possible intervals for some countries drops to two. If we select among these those with at least three intervals, we will have the countries reported in Table 4.

In this way, according to the number of time passages that the virus carries out in each country, counting the number of infected, we get pairs, triads or quaterns of differences, which give us a measure of how the virus evolves in each country. These values for each country identify the temporal evolution of the virus with respect to specific peculiarities of the viral behaviour. If we analyse these rhythms, we find that some countries show recurring or slightly modified triads. In particular, the countries are listed in Table 5.

**Table 3. Temporal intervals for the most infected countries.**

| Country | Interval I | Interval II | Interval III | Interval IV |
|---|---|---|---|---|
| Russia | 10 | 13 | 21 | |
| United States | 8 | 8 | 8 | 32 |
| Brazil | 7 | 8 | 14 | 29 |
| United Kingdom | 9 | 12 | 21 | |
| Spain | 5 | 7 | 8 | 15 |
| Italy | 2 | 6 | 10 | 20 |
| France | 8 | 11 | 19 | |
| Germany | 7 | 10 | 18 | |
| India | 10 | 15 | 15 | 35 |
| Turkey | 3 | 3 | 8 | 24 |
| Iran | 5 | 5 | 10 | 55 |
| Peru | 7 | 14 | 14 | 36 |

The steps between particularly significant dates (or days) that we have considered essential to analyse the temporal trend of the virus in each country (beginning, hot spot ignition, turning point of the pandemic trend) are calculated by counting the differences between the days in the temporal succession that identify these evolutionary points of the virus behaviour.

To further explore more significant relations between these triads, we created a graph with neighbourly connections between these rhythms. In the case of the 8 countries, with four intervals we will have the graph reported in Fig 10.

The graph with the 42 countries that have three ranges of intervals may be more significant. For the construction of this graph, we have combined a vertex with the three nearest vertexes. By using the representation of the complex graph, it is possible to adopt all the typical metrics of these systems to obtain further information, such as the centrality of one vertex with respect to the others. From these analyses, we obtained that the central vertices are {"France", "Germany", "India", "Ireland", "Israel"}, considering the central triads with a Euclidean distance. If we consider instead one vertex, with only two neighbours, the graph disconnects (Fig 11).

Let us now consider the 49 countries that have at least 2 intervals, after the second substantial step towards the spread of contagion has occurred. For these countries, we observe that 2 intervals are repeated 3 times ({6, 12} and {6, 9}) and 3 intervals are repeated twice ({8, 9}, {7, 10}, {7, 8}).

We complete this analysis by observing the distribution of these pairs in the ideal space of the rhythms that these countries have achieved with respect to the temporal development of the virus. The 3D plot in Fig 12 shows that these pairs tend to group together in a well-defined space, identifying temporal shared spaces among several countries.

In this space, 30 pairs belonging to 30 countries out of 49 countries place the first element of the interval between 5 and 10, while the second element of the interval between interval between 0–5 and 0–10, while the second element of the interval between 0–5 and 10–20, highlighting a non-random distribution of epidemic spread worldwide.

Temporal data allowed us to have the incubation times, the pandemic outbreak and the turning points of the pandemic evolution. In this way, we created the analogy with the physical systems. In particular, we detected three fundamental phases of transitions (Fig 13):

**a. Incubation or latency of the virus in a certain area of the world.** At this stage, the virus is replicating and adapting to the new environment and hosts it finds in different parts of the world. Always in analogy with physical systems, in latency there could be an accumulation of energy (viral strength? replication of the virus so that it can perform its primary function

**Table 4. Temporal intervals for 42 countries.**

| Country | Interval I | Interval II | Interval III | Interval IV |
|---|---|---|---|---|
| Russia | 10 | 13 | 21 | |
| United States | 8 | 8 | 8 | 32 |
| Brazil | 7 | 8 | 14 | 29 |
| United Kingdom | 9 | 12 | 21 | |
| Spain | 5 | 7 | 8 | 15 |
| Italy | 2 | 6 | 10 | 20 |
| France | 8 | 11 | 19 | |
| Germany | 7 | 10 | 18 | |
| India | 10 | 15 | 15 | 35 |
| Turkey | 3 | 3 | 8 | 24 |
| Iran | 5 | 5 | 10 | 55 |
| Peru | 7 | 14 | 14 | 36 |
| Saudi Arabia | 6 | 12 | 25 | |
| Chile | 6 | 9 | 25 | |
| China | 0 | 3 | 7 | |
| Pakistan | 6 | 9 | 28 | |
| Belgium | 3 | 10 | 13 | |
| Mexico | 7 | 12 | 23 | |
| Qatar | 3 | 23 | 23 | |
| Netherlands | 5 | 9 | 14 | |
| Sweden | 6 | 9 | 27 | |
| Portugal | 7 | 7 | 15 | |
| Switzerland | 5 | 8 | 12 | |
| South Africa | 7 | 9 | 44 | |
| Ireland | 8 | 9 | 21 | |
| Indonesia | 6 | 12 | 34 | |
| Kuwait | 18 | 28 | 31 | |
| Colombia | 6 | 13 | 37 | |
| Poland | 6 | 11 | 28 | |
| Ukraine | 8 | 9 | 27 | |
| Egypt | 8 | 21 | 38 | |
| Romania | 6 | 12 | 28 | |
| Israel | 10 | 11 | 19 | |
| Japan | 22 | 29 | 28 | |
| Austria | 7 | 8 | 15 | |
| Dominican Republic | 7 | 10 | 40 | |
| Philippines | 6 | 14 | 39 | |
| Afghanistan | 13 | 24 | 34 | |
| Panama | 7 | 12 | 51 | |
| Denmark | 6 | 8 | 50 | |
| Serbia | 8 | 13 | 38 | |
| Bahrain | 14 | 32 | 47 | |

This table reports the temporal intervals for the 42 considered countries.

**Table 5. Slightly modified or shared rhythms among countries.**

| Saudi Arabia | {6, 12, 25} | Chile | {6, 9, 25} |
|---|---|---|---|
| Indonesia | {6, 12, 34} | Pakistan | {6, 9, 28} |
| Romania | {6, 12, 28} | Sweden | {6, 9, 27} |
| Mexico | {7, 12, 23} | Germany | {7, 10, 18} |
| Panama | {7, 12, 51} | Dominican Republic | {7, 10, 40} |
| Ireland | {8, 9, 21} | Brazil | {7, 8, 14, 29} |
| Ukraine | {8, 9, 27} | Austria | {7, 8, 15} |

Recurring or slightly modified triads of shared worldwide rhythms among different countries.

which is to infect as many hosts as possible. We assume that the virus has two functions, to expand in space and to infect. The latency may have a state that physicists call mixed state, similar to when the water is about to boil, so not all the action of the virus is in the same way, in some parts of the world the virus has a more exuberant behaviour than in other parts of the world. This would explain the extreme difference in the outbreak of the pandemic in the various countries of the world.

**b. Pandemic outbreak.** The virus is proceeding at a very fast pace and is spreading rapidly throughout the world, adopting different times in different countries. Then the pandemic explosion begins, the points on the globe where the explosion begins are identified. At this stage, the pandemic shows the most sustained processes of contagion. The virus is in full activity. Even in this phase, the virus reaches critical phases, also presenting an oscillatory behaviour. In particular, if in the early stage, the pandemic spread by direct transmission, giving rise to an exponential growth rate [30], then other factors played an important role in the transmissions, analogously to phase transitions in clusters [31]. Spatio-temporal power law regimes may be linked to these phenomena.

**c. Control of the virus evolution, through measures of containment and social restriction.** The course of the virus and the force of contagion through spacing is slowing down. This process also slows down differently in different countries around the world. In this phase of control, there are real turbulences or critical phenomena that are chaotic as well [32] or oscillatory [33].

## 4.5 A Machine Learning to analyse rhythms

We can further investigate the data related to the pairs of intervals by building an unsupervised Machine Learning and asking it to examine the presence of emerging clusters. After several rounds of analysis, the main result is the emergence of three families composed respectively of 38, 6 and 3 pairs, whose visual representation is shown in Fig 14.

This result confirms what had already been highlighted by the histogram shown in Fig 12.

Countries that therefore have the same intervals in the dynamic development of the pandemic, albeit with slight differences, are listed in Table 6.

## 5. Methods

In what follows, we describe the methodological steps we followed to collect the data, extract the time variables, analyse the results.

Starting time: 22 January 2020. Collection of all available data from this date, for all possible countries. May 28, 2020 is the last day considered for this study; 120 is the total number of days considered to analyse the pandemic spreading. Data have been retrieved from [34]. They

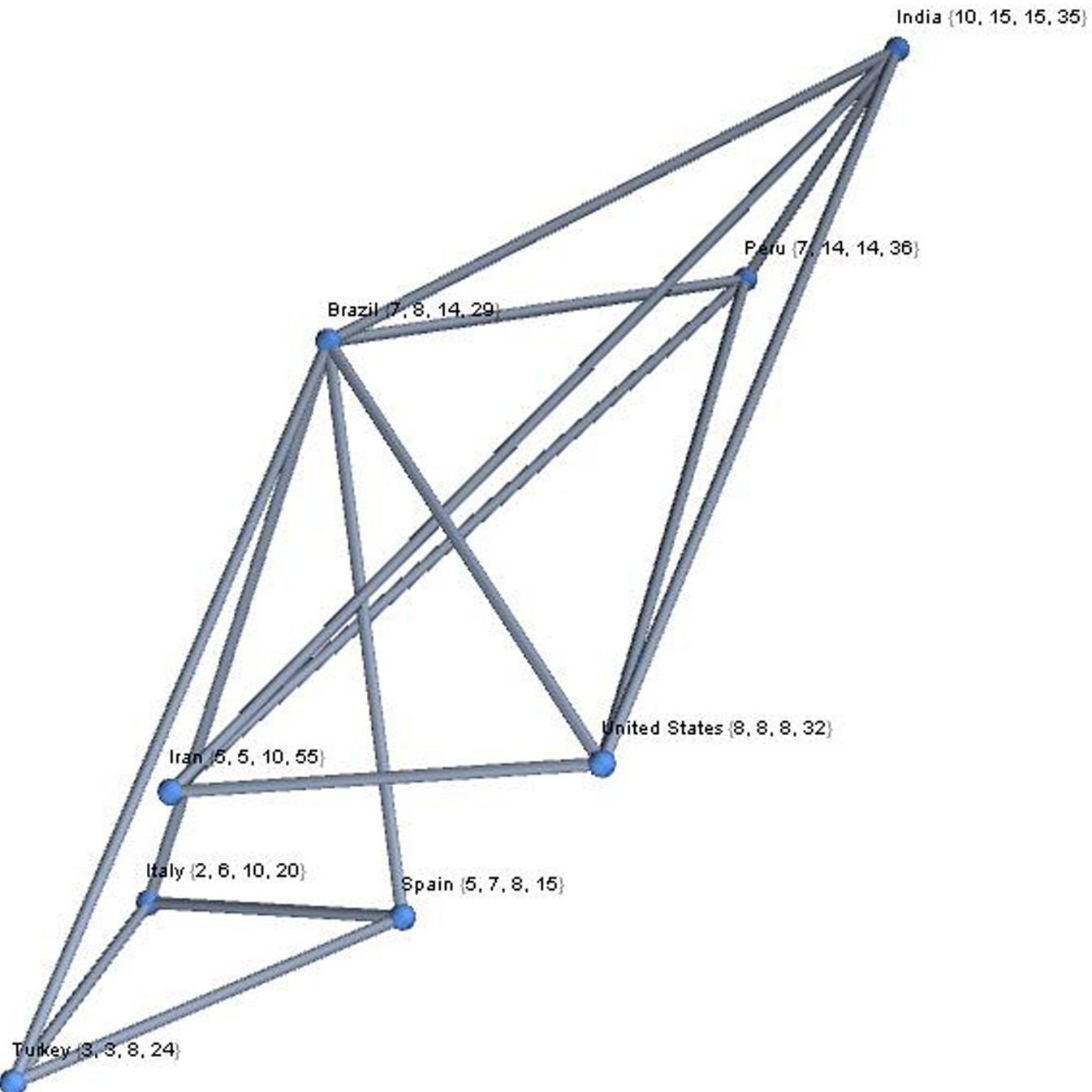

**Fig 10. Complex graph representation of the eight countries, with four temporal intervals.** This representation allows the highlighting of temporal relationships on the pandemic evolution between countries, by using complex networks statistics.

refer to estimated cases of novel coronavirus (COVID-19, formerly known as 2019-nCoV) infection by country or region. "Data is based on WHO, U.S. CDC, ECDC, China CDC (CCDC), NHC and DXY. Data for Hong Kong SAR, Macau SAR, and Taiwan have been downloaded from [35]. Data have been cleaned and aggregated by regions and countries, then harmonised by removing insecure data that did not correspond to international standard.217

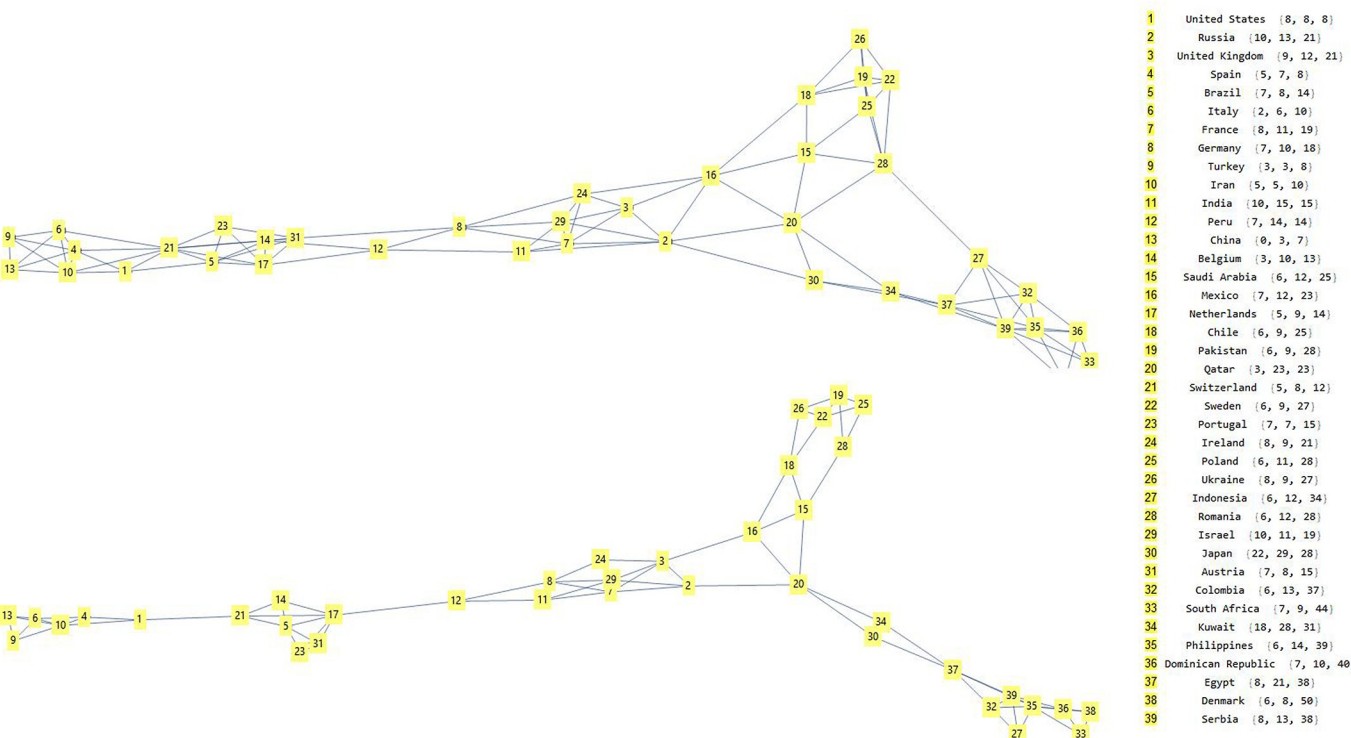

**Fig 11. Complex graph representations of the 39 countries, with different temporal interval.** In Fig 11(a), it is possible to see a graph, with the names of the 39 countries, by considering 4 neighbouring countries; in Fig 11(b), a disconnected graph, when instead we consider 3 neighbouring countries. This demonstrate how countries can be connected together. This method can be useful to detect mutual relationships among countries that share similar temporal intervals.

countries for which at least one case has occurred and the case of the cruise ship have been considered. In particular, the list is as follows:

- 217 countries + one cruise ship not associated with any country with at least one infected person,

- 210 nations (and a ship) with more than 10 cases,

- 174 nations (and ship) with more than 100 cases,

- 115 countries with more than 1,000 cases,

- 50 countries with more than 10,000 cases,

- 12 countries with more than 100,000 cases,

- 1 country with more than 1,000,000 cases.

Provided that each parameter of the compartmentalised models (SIR and SEIR) can be used, the parameters considered to calculate these epidemic courses are the following:

- t—Time (in days and in weeks, according to the representation needs),

- N—Population size,

- I(t)—number of infected at time t.

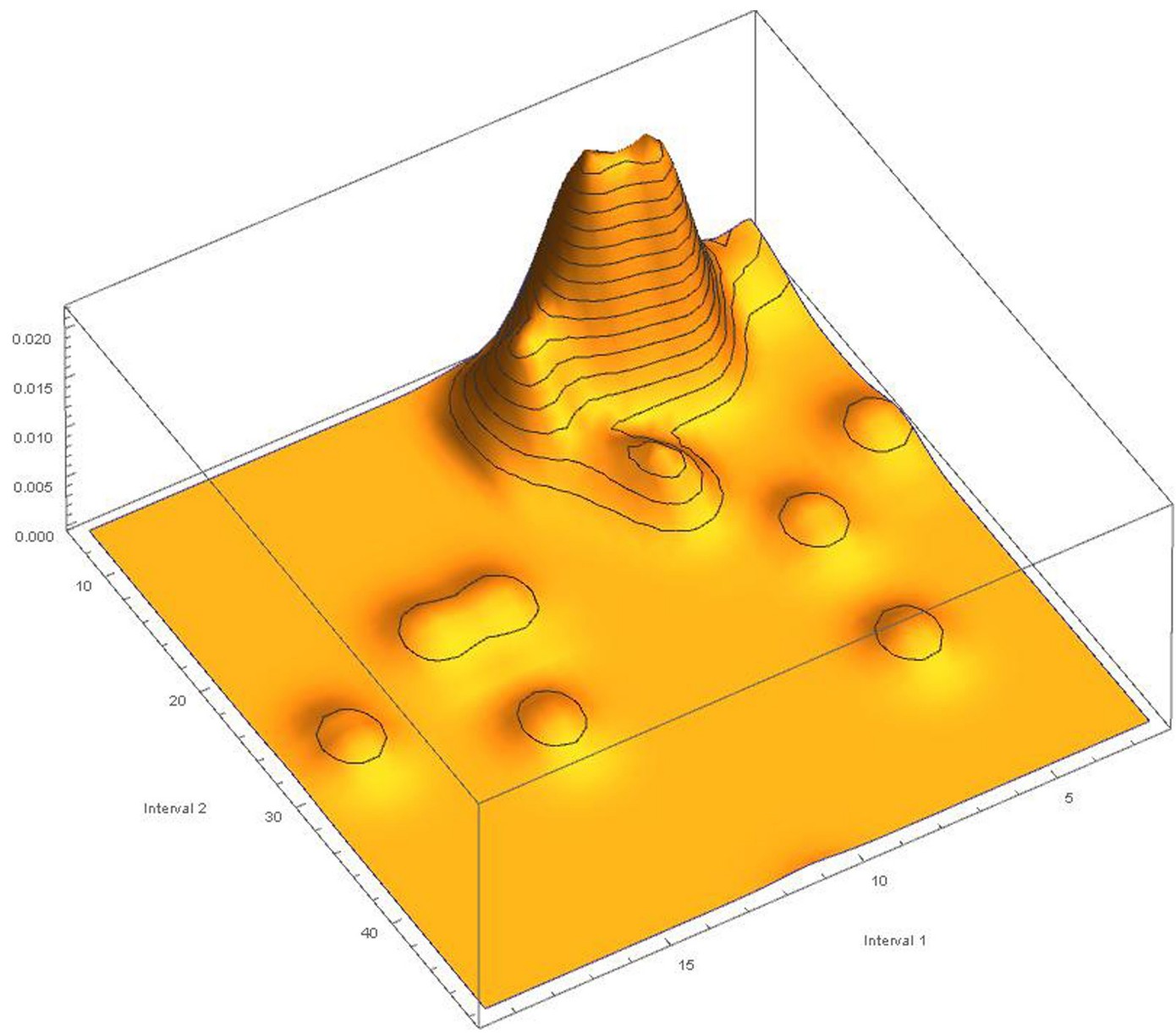

**Fig 12. Histogram of the clustering of three pairs of intervals that have the same pandemic spreading times.** Such pairs tend to group mainly in a well-defined shared temporal space.

To process the data we have used the Mathematica scientific calculation system [36] that contains a series of useful methods and tools. In particular, the following functions have been used.

- GeoListPlot to represent nations on the map using the "Mercator" method. GeoListPlot together with GeoGraphics have been used to represent the geodetic lines showing the virus dynamics on a geographical map.

- LinearModelFit to thicken the data of infected individuals on a logarithmic scale, thus validating the distributions in terms of exponential law.

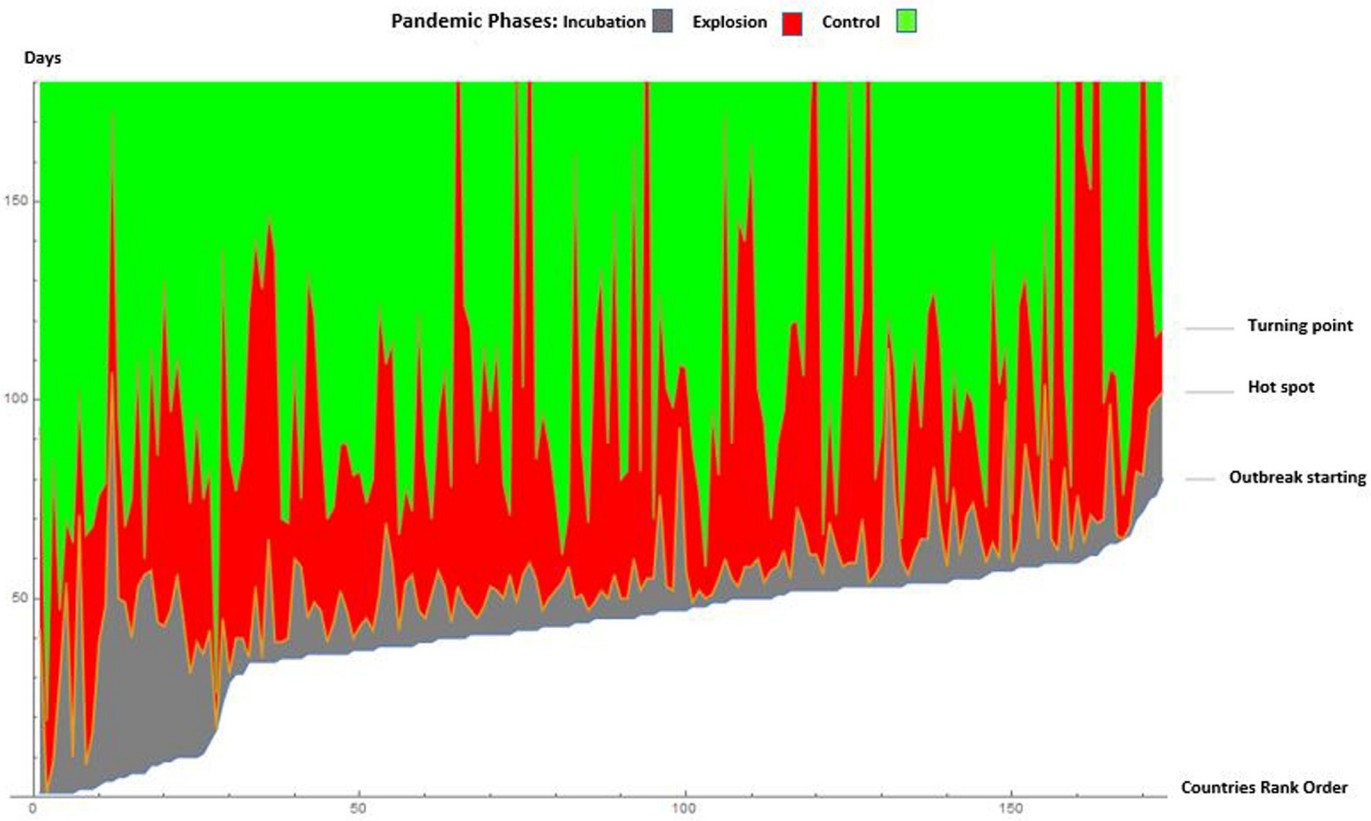

**Fig 13. Phase changes caused by the evolution of the virus.** In black the incubation phase, in red the explosion phase of the infected and in blue the control phase. The borders between the regions represent the date of the first infection, the second the date of the beginning of the hot spot and the third the date of the turning point, referred to each country.

- EstimatedDistribution on data to create the Zipf distribution.

- SmoothHistogram3D to create smooth histograms from a two-dimensional vector distribution. This function operates using the smooth kernel density estimating method.

- DateListPlot to construct time series plots with abscissae showing the days of events.

- In determining the Turning Point, we used the algorithms FindFit, FindRoot and Interpolation to improve the parameters of the interpolating function. This has allowed us to improve the results that the NonlinearModelFit that is a function that provides a nonlinear fitting of data.

- NearestNeighborGraph to build graphs between vectors of 4, 3 and 2 dimensions. The method used is selected directly from the system. In this case, the method used is the Euclidean Distance.

- FindCluster to cluster vectors. The method used is the one provided automatically by the system.

## 6. Conclusions

We analysed the most significant elements of the SARS-CoV-2 dynamics, considering: (a) the official confirmation of almost one infected person in each country, for all countries; (b) the

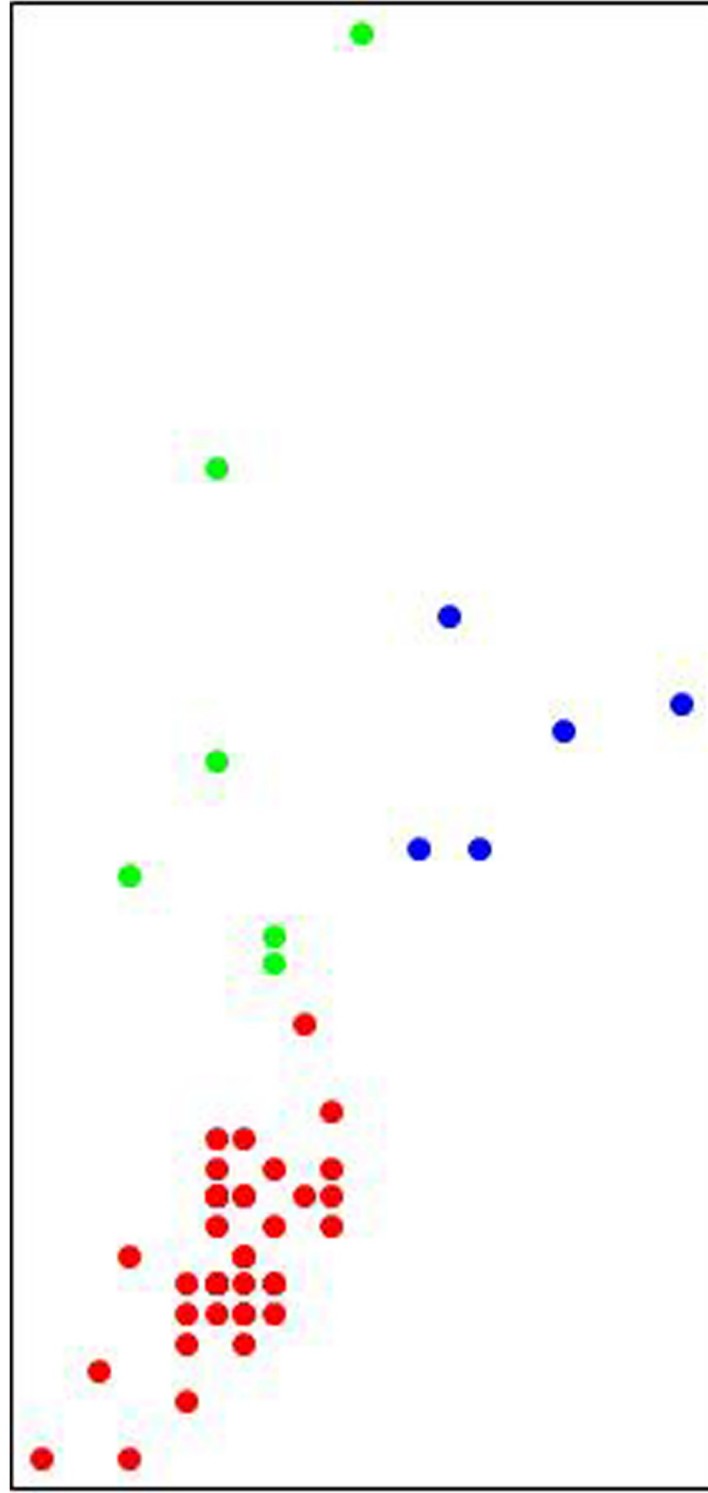

**Fig 14. Visual results of the Machine Learning algorithm.** As it is possible to note, the big family of 36 pairs of intervals is clustered together (red dots), both the families of 6 and of 3 pairs of intervals are distributed (green and blue dots respectively).

**Table 6. Countries rhythms classification by means of a Machine Learning.**

| |
| --- |
| Austria, Belarus, Belgium, Brazil, Canada, Chile, China, Colombia, Denmark, Dominican Republic, France, Germany, India, Indonesia, Iran, Ireland, Israel, Italy, Mexico, Netherlands, Pakistan, Panama, Peru, Philippines, Poland, Portugal, Romania, Russia, Saudi Arabia, Serbia, South Africa, Spain, Sweden, Switzerland, Turkey, Ukraine, United Kingdom, United States |
| Argentina, Bangladesh, Ecuador, Egypt, Qatar, South Korea |
| Afghanistan, Bahrain, Japan, Kuwait, United Arab Emirates |

Results of the ML approach. The algorithm has divided all the countries into three families. The rows contain the countries that share the same intervals.

hot spots ignition all over the world; and (c) the slowdown of the pandemic curve. Considering the time course of the infection, we have been able to notice that after the initial infection, each nation may have a latency period, to which an infectious period follows, thus allowing to the SARS-CoV-2 to adapt to different environment all over the world. We conventionally chose to count the scale steps with respect to the number of infected people per country. Any other compartmentalized variable can be used as well. Such temporal trends of the global pandemic follow an exponential law. We have extracted the rhythms of the virus evolution, obtaining for each country temporal data. In considering these temporal data, we then realized that each country is achieving not only a rate of evolution at world level, but also at the level of individual regions or provinces. In this way, we hypothesize that the dynamics that take place at a global level are in some way reproduced in the dynamics that takes place at the level of a specific country, defining areas that in turn have different rates of growth. In this article, we have analysed the growth rates of the pandemic at the global level. These results become interesting when describing a probability distribution, i.e. the probability of the occurrence of events. Consequently, the data that we have detected give us the probability distributions of the number of infected people in a certain area. It also provides us with the probability, that in the temporal passages that the virus carries out, these probability distributions are detected according to times. Therefore, by optimizing the processes that we have reported, we could build a system that, given the timing of the fundamental points we have analysed in the course of the pandemic (starting point, hot spots ignition, and slowdown of the contagion) can allow us to make predictions on the contagion dynamics. A big problem remains (along with many others that obviously need to be further analysed): the latency time of the virus before the spread of contagion develops in a massive way. This latency time suggests that the virus may not always spread immediately. The latency times give us a precise map for each country of how long it took the virus to penetrate a given ecological environment and infect human hosts. Further development suggests an adaptation of the virus to both the human host and the particular ecological-geographical environment in which the hosts live. The results have provided us with quantitative metrics of the pandemic's evolution not considered so far. The biological significance of the problem under examination regards:

1. having clearly highlighted the problem of the temporal development of the virus in the various countries of the world.

2. the distribution of the number of infected following an exponential law.

3. the temporal development in the various countries of the process of adaptation (latency) of the virus, the sudden spreading of contagion, the decreasing curve for each country, for all countries.

4. the detection of analogies with physical systems that show clear transition steps in the temporal evolution of the system and the temporal structure of viral behaviour, as key factors that give us information on the large scale dynamics of this almost completely unknown virus.

These elements not only advance the knowledge of the field, but also give the possibility to make predictions on the dynamics of contagion that can be used as customized prevention systems by different countries.

## Supporting information

**S1 File. Starting time.**
(XLSX)

**S2 File. Times.**
(XLSX)

**S3 File. Hotspot starting.**
(XLSX)

**S4 File. Intervals.**
(XLSX)

**S5 File. Turning points.**
(XLSX)

## Author Contributions

**Conceptualization:** Francesca Bertacchini, Eleonora Bilotta, Pietro S. Pantano.

**Data curation:** Francesca Bertacchini, Eleonora Bilotta, Pietro S. Pantano.

**Formal analysis:** Eleonora Bilotta, Pietro S. Pantano.

**Investigation:** Francesca Bertacchini, Pietro S. Pantano.

**Methodology:** Francesca Bertacchini, Eleonora Bilotta, Pietro S. Pantano.

**Software:** Francesca Bertacchini, Eleonora Bilotta, Pietro S. Pantano.

**Supervision:** Francesca Bertacchini, Eleonora Bilotta.

**Validation:** Francesca Bertacchini.

**Visualization:** Francesca Bertacchini, Pietro S. Pantano.

**Writing – original draft:** Francesca Bertacchini, Eleonora Bilotta, Pietro S. Pantano.

**Writing – review & editing:** Francesca Bertacchini, Eleonora Bilotta, Pietro S. Pantano.

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
