## [Decision Letter · Decision Letter 0]

26 Aug 2020

PONE-D-20-22923

On the temporal spreading of the SARS-CoV-2

PLOS ONE

Dear Dr. Bilotta Eleonora,

Thank you for submitting your manuscript to PLOS ONE. After careful consideration, we feel that it has merit but does not fully meet PLOS ONE’s publication criteria as it currently stands. Therefore, we invite you to submit a revised version of the manuscript that addresses the points raised during the review process.

We look forward to receiving your revised manuscript.

Kind regards,

Francesco Di Gennaro

Academic Editor

PLOS ONE

Journal Requirements:

Additional Editor Comments:

Dear Authors,

Follow reviewer suggestion you can improve your manuscript

I appreciate a lot your article and suggest minor revisions

Reviewers' comments:

Reviewer's Responses to Questions

**Comments to the Author**

1. Is the manuscript technically sound, and do the data support the conclusions?

Reviewer #1: Partly

Reviewer #2: Yes

2. Has the statistical analysis been performed appropriately and rigorously? 

Reviewer #1: Yes

Reviewer #2: Yes

3. Have the authors made all data underlying the findings in their manuscript fully available?

Reviewer #1: Yes

Reviewer #2: Yes

4. Is the manuscript presented in an intelligible fashion and written in standard English?

Reviewer #1: Yes

Reviewer #2: Yes

5. Review Comments to the Author

Reviewer #1: In the paper under review the authors analyze the time-space dynamics of the spread of the COVID19.

After the Introduction, in Section 2 they focus on the spatial structure of the virus diffusion. In Section 3 they pass to analyze the temporal evolution of the contagion, while in Section 4 they study the transition phases of the time evolution.

The methods they use are essentially of statistical nature. For this reason I think that the paper under review belongs to the field of statistical modeling rather than mathematical modeling, and I suggest to change the respective keyword.

What follows is a list of comments and requests of clarification that the authors should address

1. The Figure 1 is not clear to me; in particular the author should clarify how they rank the countries; this concept is used several times in the paper.

2. In several instances the authors seem to use the concept of power law, while it seems to me that the data follow an exponential decay law.

3. In Section 4.3 the authors say that the data follows the Fermi-Dirac distribution. To me it seems that the formula they write is a logistic law.

4. In Section 4.4 they say that the data display an oscillatory behavior but I could not find a full support of this statement.

5. The authors say that they have revealed a chaotic behavior of the contagion which, I think, it is not fully justified. The more generic complex behavior seems to me more appropriate.

Reviewer #2: Most of the mathematical modeling I am familiar with affords the study of the contagion following the spreading of the COVID-19 disease using a dynamical system approach, essentially based on SIR-type models. The paper under review takes an original perspective, with the aim to reveal interesting spatio-temporal features of the disease not emerged so far.

In the attached file, it follows a list of comments and questions the authors should address to improve the quality of the presentation the paper.

6. PLOS authors have the option to publish the peer review history of their article (what does this mean?). If published, this will include your full peer review and any attached files.

Reviewer #1: No

Reviewer #2: No

---

## [Author Response · Author response to Decision Letter 0]

14 Sep 2020

Dear Editor,

First, we would like to thank you and the Reviewers who have given a valuable contribution to the improvement of this work. 

We have addressed and solved some problems that the reviewers have suggested. In particular, changes have been made to improve some important technical issues on the evolution of the SARS-CoV-2 pandemic, which have certainly provided the work with precision and greater depth from a formal point of view.

After completion of the suggested edits, the revised manuscript has benefitted from an improvement in the overall presentation and clarity.

Therefore, the work is much more accurate. 

We would like to thank again the valuable work of the reviewers who have carefully read the work and found the problematic points. We are waiting for feedbacks from them, hoping they will be positive.

---

## [Decision Letter · Decision Letter 1]

5 Oct 2020

On the temporal spreading of the SARS-CoV-2

PONE-D-20-22923R1

Dear Dr. Bilotta,

We’re pleased to inform you that your manuscript has been judged scientifically suitable for publication and will be formally accepted for publication once it meets all outstanding technical requirements.

Kind regards,

Francesco Di Gennaro

Academic Editor

PLOS ONE

Additional Editor Comments (optional):

Dear Authors, congratulations!

Reviewers' comments:

Reviewer's Responses to Questions

**Comments to the Author**

1. If the authors have adequately addressed your comments raised in a previous round of review and you feel that this manuscript is now acceptable for publication, you may indicate that here to bypass the “Comments to the Author” section, enter your conflict of interest statement in the “Confidential to Editor” section, and submit your "Accept" recommendation.

Reviewer #1: (No Response)

Reviewer #2: All comments have been addressed

2. Is the manuscript technically sound, and do the data support the conclusions?

Reviewer #1: (No Response)

Reviewer #2: Yes

3. Has the statistical analysis been performed appropriately and rigorously? 

Reviewer #1: (No Response)

Reviewer #2: Yes

4. Have the authors made all data underlying the findings in their manuscript fully available?

Reviewer #1: (No Response)

Reviewer #2: Yes

5. Is the manuscript presented in an intelligible fashion and written in standard English?

Reviewer #1: (No Response)

Reviewer #2: Yes

6. Review Comments to the Author

Reviewer #1: (No Response)

Reviewer #2: (No Response)

7. PLOS authors have the option to publish the peer review history of their article (what does this mean?). If published, this will include your full peer review and any attached files.

Reviewer #1: No

Reviewer #2: No

---

## [Editor Report · Acceptance letter]

9 Oct 2020

PONE-D-20-22923R1 

On the temporal spreading of the SARS-CoV-2 

Dear Dr. Bilotta:

I'm pleased to inform you that your manuscript has been deemed suitable for publication in PLOS ONE. Congratulations! Your manuscript is now with our production department. 

Kind regards, 

on behalf of

Dr. Francesco Di Gennaro 

Academic Editor

PLOS ONE